# Calreticulin Regulates β1-Integrin mRNA Stability in PC-3 Prostate Cancer Cells

**DOI:** 10.3390/biomedicines10030646

**Published:** 2022-03-11

**Authors:** Yueh-Chien Lin, Yuan-Li Huang, Ming-Hua Wang, Chih-Yu Chen, Wei-Min Chen, Yi-Cheng Weng, Pei-Yi Wu

**Affiliations:** 1Vascular Biology Program, Department of Surgery, Boston Children’s Hospital, Harvard Medical School, Boston, MA 02115, USA; linboss87@gmail.com; 2Department of Medical Laboratory Science and Biotechnology, Asia University, Taichung 41354, Taiwan; yuanli@asia.edu.tw; 3Department of Medical Research, China Medical University Hospital, China Medical University, Taichung 41354, Taiwan; 4Department of Radiation Oncology, Lianxin International Hospital, Taoyuan 32001, Taiwan; thantkyaw.yu@gmail.com; 5Department of Life Science, National Taiwan University, Taipei 10617, Taiwan; a0939897580@gmail.com (C.-Y.C.); j810717@gmail.com (W.-M.C.); alex05612@gmail.com (Y.-C.W.); 6Department of Radiation Oncology, University of Texas Southwestern Medical Center, Dallas, TX 75390, USA; 7Department of Life Sciences, National Central University, Taoyuan 32001, Taiwan

**Keywords:** calreticulin, integrin, mRNA stability, AU-rich element

## Abstract

Prostate cancer (PCa) is the major cause of cancer-related death among aging men worldwide. Recent studies have suggested that calreticulin (CRT), a multifunctional chaperon protein, may play an important role in the regulation of PCa tumorigenesis and progression. However, the underlying mechanisms are still unclear. Integrin is an important regulator of cancer metastasis. Our previous study demonstrated that in J82 bladder cancer cells, CRT affects integrin activity through FUBP-1-FUT-1-dependent fucosylation, rather than directly affecting the expression of β1-integrin itself. However, whether this regulatory mechanism is conserved among different cell types remains to be determined. Herein, we attempted to determine the effects of CRT on β1-integrin in human prostate cancer PC-3 cells. CRT expression was suppressed in PC-3 cells through siRNA treatment, and then the expression levels of FUT-1 and β1-integrin were monitored through RT-PCR. We found that knockdown of CRT expression in PC-3 cells significantly affected the expression of β1-integrin itself. In addition, the lower expression level of β1-integrin was due to affecting the mRNA stability. In contrast, FUT-1 expression level was not affected by knockdown of CRT. These results strongly suggested that CRT regulates cellular behavior differently in different cell types. We further confirmed that CRT directly binds to the 3′UTR of β1-integrin mRNA by EMSA and therefore affects its stability. The suppression of CRT expression also affects PC-3 cell adhesion to type I collagen substrate. In addition, the levels of total and activated β1-integrin expressed on cell surface were both significantly suppressed by CRT knockdown. Furthermore, the intracellular distribution of β1-integrin was also affected by lowering the expression of CRT. This change in distribution is not lysosomal nor proteosomal pathway-dependent. The treatment of fucosydase significantly affected the activation of surface β1-integrin, which is conserved among different cell types. These results suggested that CRT affects the expression of β1-integrin through distinct regulatory mechanisms.

## 1. Introduction

Prostate cancer (PCa) is one of the most common cancers and is the major cause of cancer-related death among aging men worldwide. Statistically, about one out of every 38 men dies from the disease [1,2]. PCa tumors usually originate from the glandular tissue of the prostate with uncontrolled proliferation. Androgen has been known to play a vital role in the progression of PCa [3,4,5,6]. Therefore, hormone therapy, androgen deprivation or androgen ablation, has remained a standard therapy for patients with metastatic PCa for the past several decades [4]. However, hormone therapy is not curative, and the cancer frequently relapses to a more malignant form. In order to elucidate the mechanism by which PCa relapse from the androgen deprivation therapy, the androgen-responsive genes were investigated extensively. Calreticulin (CRT) is one androgen-responsive gene in prostatic epithelial cells [7]. It has been demonstrated that both mRNA and protein expression level of CRT are downregulated by castration and upregulated by androgen replacement in the prostate. In addition, the expression level of CRT in the prostate is much higher than that in seminal vesicles, heart, brain, muscle, kidney, and liver. This evidence demonstrates that CRT may play an important role in the androgen-related response, including the regulation of PCa tumorigenesis and progression.

Calreticulin is a multifunctional chaperon protein and participates in a variety of important biological processes. As a molecular chaperone, CRT functions to ensure proper folding of glycoproteins. CRT also possesses high binding affinity to Ca^2+^ and is involved in the modulation of intracellular Ca^2+^ homeostasis. In addition, several studies have demonstrated that CRT mediates integrin activation, which regulates cell adhesion and tumor cell metastasis [8,9,10]. Recently, several studies further suggest that CRT is an RNA-binding protein and mediates the regulation of RNA stability. In vascular smooth muscle cells, serine dephosphorylation and tyrosine phosphorylation of CRT promoted its binding with AU-rich elements (AREs) at the 3′UTR of mRNA of angiotensin type 1 (AT1) receptor, which increases mRNA stability of AT1 receptor [11]. Under high-glucose conditions, CRT destabilizes glucose transporter-1 (GLUT-1) mRNA in vascular endothelial and smooth muscle cells [12].

Integrins are a family of αβ-heterodimeric transmembrane receptors involved in cell–cell and cell–matrix interactions and regulate numerous cellular functions, including cell adhesion, cell migration, tumor invasion, and metastasis [13]. Integrin complexes are composed of 18 α and 8 β subunits, and 24 distinct types of integrins were discovered [14]. They were served as receptors of ECM ligands with different affinities [14]. The functions of integrin are regulated by a wide variety of molecular events, such as glycosylation. Glycosylation is the most common post-translational modifications for proteins and lipids [15]. Aberrant glycosylation of integrin usually leads to impaired cell–cell adhesion, activation of oncogenic signaling pathways, and induction of pro-metastatic phenotypes [16].

The previous study has demonstrated that CRT stabilized the mRNA of fucosyltransferase1 (FUT1), resulting in the activation of β1-integrin by fucosylation, thereby enhancing cell adhesion and cell migration in bladder cancer cells [10]. However, the effects of CRT on the regulation of β1-integrin and the underlying mechanism in PCa cells remains elusive. In this study, PC-3 cell was used as a PCa model to investigate whether β1-integrin activation or expression was regulated by CRT with its RNA-binding activity.

## 2. Materials and Methods

### 2.1. Cell Culture

PC-3 human prostate cancer cell line was purchased from American Type Culture Collection (ATCC, Manassas, VA, USA). PC-3 cells were maintained in RPMI 1640 medium (GE Healthcare Life Sciences, Chicago, IL, USA) supplemented with 10% fetal bovine serum (Gibco, Carlsbad, CA, USA) under a humidified atmosphere of 5% CO_2_ at 37 °C. For the subcultures, cells were trypsinized with 0.05% EDTA-trypsin (Life Technologies, Carlsbad, CA, USA).

### 2.2. Transfection of siRNA

The siRNAs of CRT were obtained from Santa Cruz Biotechnology (Dallas, TX, USA). The siRNAs were transfected using Lipofectamine^®^ 2000 (Invitrogen, Carlsbad, CA, USA) according to the manufacturer’s instruction.

### 2.3. Western Blot Assay

Cells were trypnized and washed once with cold PBS (pH 7.4, 137 mM NaCl, 2.7 mM KCl, 10 mM Na_2_HPO_4_ and 1.8 mM KH_2_PO_4_), and then collected by centrifugation at 2000 rpm for 5 min. Cells were lysed in lysis buffer (pH 8.0, 20 mM Tris, 150 mM NaCl, 1% NP-40, 1 mM Na_3_VO_4_, and 10% glycerol) containing 1% protease inhibitor Cocktail (Merck Millipore, Billerica, MA, USA). The cell lysates were incubated on ice for 15 min, followed by centrifugation at 13,000 rpm for 15 min. The supernatants were collected, and the protein concentrations were measured using Bradford assay (Bio-Rad, Hercules, CA, USA) according to the manufacturer’s instructions. The cell lysates were mixed with 1/5 volume of 5× sample buffer and boiled at 100 °C for 10 min. The samples were separated by SDS/PAGE (80 V for 15 min in 4% stacking gel and 120 V for 1.5 h in 10% running gel) and transferred to polyvinylidene fluoride membranes (100 V for 90 min at 4 °C). The membrane was blocked with 5% BSA in TBST (pH 7.4 25 mM Tris, 150 mM NaCl, 2 mM KCl, and 0.1% Tween-20) for 1 h at room temperature, followed by incubation with primary antibody overnight at 4 °C with gentle oscillation. After washing three times with TBST for 10 min, the membranes were incubated with HRP-conjugated secondary antibody for 1 h at room temperature. The membranes were washed three times with TBST for 10 min, and signals were then visualized using ECL reagent (Advansta, Menlo Park, CA, USA) according to the manufacturer’s instructions. The antibodies used were as follows: rabbit polyclonal anti-CRT antibody (PA3-900, Thermo Fisher Scientific, Waltham, MA, USA), rabbit polyclonal anti-GAPDH antibody (GTX100118, Genetex, Irvine, CA, USA), rabbit polyclonal anti-VEGF-A antibody (GTX102643, Genetex, Irvine, CA, USA), rabbit polyclonal anti-ITGB1 antibody (MAB2252, Millipore, Burlington, MA, USA), rabbit polyclonal anti-HuR antibody (3A2, Santa Cruz Biotechnology, Dallas, TX, USA), and rabbit polyclonal anti-TTP antibody (ABE285, Millipore).

### 2.4. Total RNA Extraction and Reverse Transcription Quantitative PCR (RT-QPCR)

Total RNA was extracted by TRIzol reagent (Invitrogen, Carlsbad, CA, USA). The first strand cDNA was synthesized with 1 μg total RNA using a Toyobo reverse transcription (RT)-polymerase chain reaction (PCR) kit (Toyobo, Osaka, Japan). Quantitative real-time PCR reactions were conducted in a Mini-Opticon Real-Time detection system (Bio-Rad, Hercules, CA, USA) using iQ^TM^ SYBR^®^ Green Supermix (Bio-Rad, Hercules, CA, USA). The thermal profiles of real-time PCR was 95 °C for 5 min, followed by 40 cycles of 95 °C for 30 s, and 60 °C for 30 s. The melting curve of each tube was examined to confirm a single peak appearance. The sequences of paired primers for real-time PCR detection are as follows: CRT forward: 5′-CCT CCT CTT TGC GTT TCT TG-3′, CRT Reverse: 5′-CAG ACT CCA AGC CTG AGG AC-3′; β1-integrin 3-UTR forward: 5′-TGC AAC AGC TCT CAC CTA CG-3′, β1-integrin 3-UTR reverse:5′- GAT GGG CAA CTC AAA TGG TGA-3′; GAPDH forward: 5′-GGT GGT CTC CTC TGA CTT CAA C-3′, GAPDH reverse: 5′-TCT CTC TTC CTC TTG TGT TCT TG-3′.

### 2.5. Determination of mRNA Stability

Cells (1 × 10^5^) were seeded on polystyrene 6-well plate (Corning) and treated with 2.5 µg/mL actinomycin D (Act-D, Sigma, St. Louis, MO, USA) at the indicated time points. Subsequently, total RNAs were extracted and reverse-transcribed to cDNA. The real-time PCR was conducted to quantify the levels of mRNA.

### 2.6. Dual-Luciferase Reporter Assay

The full-length *β1-integrin* 3′UTR (*β1-integrin*-3UTR-FL) and truncated *β1 integrin* 3′UTR (*β1-integrin*-3UTR-truncate) were amplified from cDNA of PC-3 cells by PCR with the primers: *β1-integrin*-3UTR-FL forward: 5′-GGC TCG AGA ATG AGT ACT GCC CGT GCA AAT-3′, *β1-integrin*-3UTR-FL reverse: 5′-GGG CGG CCG CTC CGA TTT AAG TAT TTT AGG-3′, *β1-integrin*-3UTR-truncate forward: 5′-AAC TCG AGA ATG AGT ACT GCC CGT GCA AAT CC-3′, *β1-integrin*-3UTR-truncate reverse: 5′-AAG CGG CCG CAT GGC ACT AAC TCA AAG TAA-3′. The fragments were subsequently inserted into psiCHECK™-2 vector (Promega, Madison, WI, USA) between the XhoI and NotI cutting sites. The psiCHECK™-2 vector contained two reporter genes, *Renilla* and *Firefly* luciferases, and is designed for the endpoint lytic assay. *Firefly* luciferase was used to normalize *Renilla* luciferase expression.

PC-3 cells (2 × 10^4^) were seeded onto 24-well plates and cultured for 24 h prior to transfection. Cells were transfected with 625 ng plasmid DNA (psiCHECK2-β1-integrin-3UTR-FL or psiCHECK2-β1-integrin-3UTR-truncate) by Lipofectamine^®^ 3000. Four hours after transfection, the medium was replaced with normal culture medium, and cells were cultured for another 48 h. Luciferase assays were performed using the Dual-Glo Luciferase Reporter Assay kit (E2920, Promega, Madison, WI, USA) according to the manufacturer’s procedure. Briefly, cells were lysed with 100 µL Glo lysis buffer (E2661, Promega, Madison, WI, USA). The cell lysates in the plate were oscillated on the orbital shaker for 10 min at room temperature and then transferred into Eppendorf tubes. For each reaction, 25 µL cell lysate was added into a 96-well white plate and mixed with 25 µL Dual-Glo^®^ Reagent followed by incubation for 10 min. The *Firefly* luciferase activity was first recorded using the SpectraMax M5 (Molecular Devices, San Jose, CA, USA). Then, 25 µL Dual-Glo^®^ Stop & Glo^®^ substrate was added, followed by recording of *Renilla* luciferase activity.

### 2.7. RNA Immunoprecipitation (RNA-IP)

Cells were lysed by ice-cold RNA-IP lysis buffer (150 mM KCl, pH 7.4 25 mM Tris, 5 mM EDTA and 0.35% Triton X-100) containing 1× protease inhibitor cocktail. Protein concentrations were determined using Bradford assay (Bio-Rad, Hercules, CA, USA) according to the manufacturer’s instructions. Lysates containing 400 µg total protein were pre-cleaned with PureProteome™ Protein A/G Magnetic Beads (Invitrogen, Waltham, MA, USA) at 4 °C for 1 h. The pre-cleaned lysates were incubated with rabbit anti-CRT antibody (PA3-900, Thermo Fisher Scientific, Waltham, MA, USA) at 4 °C overnight. The protein–antibody complex was pulled down by magnetic beads at 4 °C for 4 h. Then, all the beads were collected and washed with ice-cold RNA-IP lysis buffer for three times, followed by three washes with NT_2_ buffer (50 mM pH 7.4 Tris, 150 mM NaCl and 0.05% NP-40). For Western blot, the beads were mixed with sample buffer and then boiled at 100 °C for 15 min for protein elution. For real-time PCR, beads were subjected to RNA extraction by TRIzol reagent.

### 2.8. RNA Electrophoretic Mobility Shift Assay (RNA EMSA)

Biotinylated RNA probes for β1-integrin-ARE were prepared using in vitro transcription. Briefly, the linearized PCR fragments containing T7 promotor sequence were amplified from psiCHECK2-β1-integrin-3UTR-FL plasmid by PCR. The sequences of the paired primers were as follows: β1-integrin ARE-probe forward: 5′-TAA TAC GAC TCA CTA TAG GGA TAC TGT GGC TAT GCA ACA G-3′, β1-integrin ARE-probe reverse: 5′-CAT CAG AGT CAA GAC ATC CGA T-3′. The PCR fragments were transcribed to RNA by the incubation of T7 polymerases (Promega, Madison, WI, USA) at 37 °C for 4 h, followed by incubation of Recombinant DNase I (Promega, Madison, WI, USA) at 37 °C for 1 h. The transcribed RNAs were purified by phenol–chloroform extraction and resuspended in DEPC-treated water. Biotin-UTP were then added on the 3′-end of RNA probes using Pierce™ RNA 3′ End Biotinylation Kit (Thermo Fisher Scientific, Waltham, MA, USA) according to the manufacturer’s instruction.

Cell lysates from PC-3 cells were prepared for EMSA. The cells were lysed with ice-cold All Purpose Buffer (APB, 50 mM Tris-HCl pH7.5, 250 mM NaCl, 3 mM EDTA, 3 mM EGTA, 1% Triton X-100, 0.5% NP-40, 10% Glycerol) containing 1× protease inhibitor cocktail (Merck Millipore, Burlington, MA, USA). The cell lysates were incubated on ice for 15 min, followed by centrifugation at 13,000 rpm for 15 min. The supernatants were collected, and the protein concentrations were determined using Bradford assay.

RNA EMSA was performed using LightShift^®^ Chemiluminescent RNA EMSA Kit (Thermo Fisher Scientific, Waltham, MA, USA) according to the manufacturer’s instructions. Briefly, 4 μg cell lysates were incubated with 10× binding buffer at room temperature for 10 min. After heating at 95 °C for 5 min, the RNA probes were gradually cooled on ice, followed by adding the biotinylated RNA probes into mixture and incubating at room temperature for 20 min. Then, 5 μL 5× loading dye was added to the mixtures. Additionally, 5% non-denaturing polyacrylamide gel containing 0.5× TBE (pH 8.3, 45 mM Tris-borate, and 1 mM EDTA) was pre-run at 100 V for 1 h, and the reaction mixtures were then separated in the non-denaturing polyacrylamide gel at 100 V for about 1.5 h. The samples were transferred to Nylon membrane at 400 mA at 4 °C for 60 min. The membrane was crosslinked at 120 mJ/cm^2^ using a UV-light crosslinking instrument equipped with 254 nm bulbs. Signals were visualized using Chemiluminescent Nucleic Acid Detection Module (Thermo Fisher Scientific, Waltham, MA, USA) according to the manufacturer’s instructions.

### 2.9. Immunocytochemistry

Cells were seeded onto poly-l-lysine pre-coated coverslips in 6-well plates. After 24 h incubation, cells were fixed with 4% paraformaldehyde and permeabilized by 0.2% Triton X-100 in PBS at room temperature for 10 min. Detergent-permeabilized cells were blocked with 3% bovine serum albumin in PBST (0.1% Tween 20 in PBS) at room temperature for 30 min and were incubated with mouse anti-β1-integrin (total β1-integrin, Clone JB1B, Santa Cruz) and rat anti-β1-integrin (active β1-integrin, Clone 9EG7, BD Pharmingen™, Franklin Lakes, NJ, USA) antibodies in PBS/1.5% bovine serum albumin at 4 °C overnight. After extensive washes with PBST, cells were incubated with an Alexa Fluor 546 donkey anti-mouse IgG and Alexa Fluor 488 donkey anti-rat IgG in PBS/1.5% bovine serum albumin 37 °C for 1 h. Images were captured by Leica TCS SP5 confocal microscope.

### 2.10. Proliferation Assay by Cell Counting

siCtrl- and siCRT-transfected PC-3 cells were seeded in the 24-well plates in the density of 5 × 10^4^ cells/well. After three days of incubation, the cells were trypsinized and stained with 0.4% Trypan Blue. The cell number was counted by the hemocytometer.

### 2.11. Cell Adhesion Assay

The 96-well culture plates were coated with type I collagen (10 μg/mL) and then incubated at 37 °C overnight, followed by a PBS wash. Cells (5 × 10^4^ cells/100 μL) were seeded in the well and incubated at 37 °C for 10–120 min. After removing the culture medium with non-attached cells, the wells were washed with PBS 3 times. Then, 0.1% crystal violet was applied to stain the attached cell for 10 min. After three PBS washes, 10% acetic acid was added for 20 min. The absorbance of cell lysate at 550 nm was measured using a spectrophotometer.

### 2.12. Wound Healing Assay

PC-3 cells (3 × 105 cells/well) were seeded in 24-well plates to grow in a monolayer for 24 h. Then, a wound was made in each well by scratching with a sterile 20–200 μL pipette tip. The detached cells were removed by PBS wash. Then, 500 μL of fresh medium was added afterward and incubated for 48 h. The scratch closure was monitored and imaged in 24 h intervals using a BioTek Cytation7 imaging system.

### 2.13. Statistical Analysis

All statistical comparisons between groups were determined using Student’s *t*-test by GraphPad Prism (San Diego, CA, USA). Each result was obtained from at least three independent experiments, and a value of * *p* < 0.05, ** *p* < 0.01, *** *p* < 0.001 was considered statistically significant.

## 3. Results

### 3.1. Knockdown of CRT Suppresses Expression Levels of β1-Integrin in PC-3 Cells

Our previous study in bladder cancer has demonstrated that CRT affects β1-integrin activity through FUBP-1-FUT-1-dependent fucosylation, rather than directly affecting the expression of β1-integrin itself [10]. To confirm whether this mechanism is conserved in PCa cells, CRT was knocked down by transfection of siRNA in PC-3 cells. Compared to the negative control cells, CRT siRNA significantly suppressed the protein expression levels of CRT, resulting in the reduction of β1-integrin (Figure 1A). This result indicated that the expression level of β1-integrin is regulated by CRT directly, which might be FUT1 independent. In addition, the mRNA expression level of β1-integrin was downregulated in CRT-knockdown (CRT-KD) PC-3 cells (Figure 1B). With the treatment of transcription inhibitor actinomycin D (Act-D), we also found that the mRNA stability of β1-integrin was significantly reduced in CRT-KD PC-3 cells. These results indicated that CRT regulates β1-integrin expression by stabilizing its mRNA.

### 3.2. Depletion of CRT Inhibits β1-Integrin Expression by Regulating mRNA Stability through AU-Rich Element

It is known that AU-rich element (ARE), enriched with adenosine and uridine nucleotides in the 3′UTR of mRNA, plays critical roles in controlling mRNA degradation. By the analysis from the ARE database (AU-RICH ELEMENT DATABASE, Available online: http://brp.kfshrc.edu.sa/ARED/ (accessed on 10 October 2016)), we found an ARE sequence located at 3′UTR of β1-integrin mRNA (Figure 2A). Therefore, we hypothesized that ARE is required for the regulation of β1-integrin mRNA stability by CRT. To examine this hypothesis, the full-length (FL) and ARE-truncated 3′UTR of β1-integrin mRNA were constructed into the dual-luciferase reporter plasmid containing *Renilla* (hRLuc) and *Firefly* (hLuc) luciferases, respectively (Figure 2A). PC-3 cells were then transfected with the constructed reporter plasmids for 48 h with or without CRT siRNA. In this system, the effect of ARE on the RNA stability can be determined by the ratio of luciferase signal of *hRLuc* to *hLuc*. The result showed that knockdown of CRT significantly decreases the ratio of luciferase activity compared to the control cells in FL-transfected cells (Figure 2B). However, no significant difference was found in the 3′UTR truncate groups. These results suggested that CRT stabilize β1-integrin mRNA stability through specifically recognizing ARE sequence.

### 3.3. CRT Indirectly Interacts with Are at β1-Integrin mRNA

To further confirm that CRT interacts with mRNA of β1-integrin through ARE sequence in PC-3 cells, an RNA-immunoprecipitation (RNA-IP) assay was performed. Total cell lysate of PC-3 was incubated with CRT antibody or normal rabbit IgG (control), and the mixture was captured by protein A/G beads. The CRT-bound RNAs were extracted by TRIzol reagent and subjected to reverse transcription followed by real-time PCR. We found that CRT could be successfully pulled down by the specific antibody (Figure 3A), and β1-integrin mRNA was detected and enriched in the pull-downed CRT binding complex (Figure 3B). By the RNA electrophoretic mobility shift *assay* (EMSA), the interaction between CRT and mRNA of β1-integrin was further investigated (Figure 3C). Adding CRT antibody results in a specific upper super-shift band (Figure 3C, Lane 3), whereas no super-shift band was found when adding non-biotin-labeled RNA competitor (Figure 3C, Lane 4). This result provides further evidence to support that CRT binds with ARE at the 3′UTR of β1-integrin mRNA. Interestingly, we also found that adding the antibody of HuR, a known ARE-binding protein, could also induce the super-shift of the band (Figure 3C, Lane 5), indicating that HuR may also be involved in the binding complex of CRT at β1-integrin mRNA. To determine whether CRT directly interacts with ARE, recombinant CRT instead of the total cell lysate was utilized in the EMSA. However, no band shift was observed upon adding the recombinant CRT (Figure 3D). Altogether, these results indicated that CRT indirectly binds to the ARE of β1-integrin mRNA, and the indirect binding might need interaction with other ARE-binding proteins, e.g., HuR.

### 3.4. Knockdown of CRT Induces the Internalization of β1-Integrin

Our current data have demonstrated that knockdown of CRT inhibits β1-integrin expression by regulating mRNA stability. To further explore the effects of CRT on β1-integrin expression, immunofluorescence staining of total and active β1-integrin in PC-3 cells was performed by specific antibodies. We found that both total and active β1-integrin expression were reduced in the CRT knockdown PC-3 cells (Figure 4A). Interestingly, we also found that the level of β1-integrin, either total or active forms, on the cell surface was significantly decreased by CRT siRNA (Figure 4A). Most of β1-integrin translocates from the cell membrane into the cytosol. This observation may suggest that CRT not only controls β1-integrin expression but also regulates its cellular distribution, which might affect the adhesion and migration of PC-3 cells. However, the underlying mechanism still needs further investigations.

### 3.5. The Internalization of β1-Integrin Is Proteasome and Lysosome Independent

To investigate whether the internalization of β1-integrin was related to its degradation by proteasome or lysosome, we treated the CRT KD PC-3 cells with proteosome or lysosome inhibitors, MG132 and Bafilomycin A1, respectively. The immunoblotting result showed that neither MG132 nor Bafilomycin A1 blocked siCRT-induced β1-integrin reduction, which indicates that the internalization of β1-integrin is proteasomal or lysosomal degradation-independent (Figure 4B). In addition, this result further confirmed that the decreased protein level of β1-integrin by knockdown of CRT is mediated by RNA stability but not by protein degradation. Here, we also observed the protein expression level of FUT1, and the result shows that knockdown of CRT did not affect the FUT1 expression. This data further provides evidence that in PCa, CRT directly regulates β1-integrin expression, but not through FUT1, which is different from the mechanism in bladder cancer.

### 3.6. Activation of β1-Integrin in PC-3 Cells Is Fucosylation Dependent

Our previous study has demonstrated that CRT affects integrin activity through FUBP-1-FUT-1-dependent fucosylation in J82 bladder cancer cells. In this study, we proposed an alternative mechanism in PCa cells that CRT regulates β1-integrin mRNA stability without change in FUT-1 expression. Thus, we would next like to investigate whether the activation of β1-integrin in PCa cells is still fucosylation-dependent. PC-3 cells were treated with fucosidase and BSA as a negative control. Total and active β1-integrin were analyzed by immunostaining (Figure 5). We found that active β1-integrin was dominantly expressed in the cell surface, which was significantly suppressed by the fucosidase treatment. The expression level of total β1-integrins was not affected by fucosidase. These results suggested that in PCa cells, the activation of β1-integrin is also regulated by a conserved fucosylation mechanism.

### 3.7. Knockdown of CRT Inhibits the Cell Proliferation, Adhesion and Migration of PCa Cells

To understand the effects of CRT on the tumor progression of PCa, PC-3 cells were transfected with CRT siRNA, and the tumor cell behaviors were analyzed in vitro. We found that knockdown of CRT inhibited the cell proliferation of PC-3 cells (Figure 6A). In addition, the ability of cell adhesion to the type I collagen was significantly reduced by CRT siRNA (Figure 6B). To test cell migration ability, a wound-healing assay was performed, and we found that wound area recovery rate was decreased in the CRT KD cells (Figure 6C). All these results suggest that CRT may play an oncogenic role in the progression of PCa. Targeting CRT could be a potential therapeutic for the PCa treatment.

## 4. Discussion

Gene expression in eukaryotic cells is highly regulated by multiple processes, such as mRNA splicing and mRNA decay. The degradation of mRNA is mediated by the interaction between RNA-binding proteins (RBPs) and mRNA structures, including 5′-capping structure, 5′- and 3′-untranslated region (UTR), and 3′-polyadenylate (polyA) [17]. AU-rich elements (AREs) in 3′UTR were considered the most common *cis*-acting factor that regulates mRNA stability in mammalian cells [18]. AREs are composed of multiple copies of the AUUUA motif, which bind directly or indirectly with RNA-binding proteins to regulate mRNA stability [18]. Previous studies have suggested that CRT is a novel RNA-binding protein involved in the regulation of RNA stability. In vascular cells under high-glucose conditions, CRT has been shown to destabilize glucose transporter-1 (GLUT-1) mRNA [12]. Dephosphorylated and phosphorylated CRT on serine and tyrosine, respectively, resulted in the binding of CRT with ARE at the 3′UTR of mRNA of angiotensin type 1 (AT1) receptor [11]. This interaction is required for the regulation of mRNA stability of AT1 receptor. In this study, we demonstrate a novel mechanism whereby CRT regulates β1-integrin expression with its RNA binding property in PCa cells. By binding to the ARE at the 3′UTR of β1-integrin mRNA, CRT increases the stability of β1-integrin mRNA, resulting in a higher expression level of β1-integrin (Figure 2 and Figure 3). This mechanism is distinct from our previous findings in bladder cancer that CRT stabilized the mRNA of FUT1 resulting in the activation of β1-integrin by fucosylation [10]. Therefore, CRT regulates β1-integrin in a manner that is dependent on the cell types. However, multiple PCa cell lines should be tested to confirm that this specialized regulation of β1-integrin mRNA stability by CRT is universal in PCa cells.

Our current data suggested that CRT regulates β1-integrin mRNA stability by indirect binding to its 3′UTR (Figure 3). That is, CRT requires other RNA-binding proteins to interact with AREs of β1-integrin mRNA. Several ARE-binding proteins, such as HuR and TTP, have been identified as regulators for mRNA stability [19]. From our EMSA results in Figure 3C, we have found that anti-HuR antibody shows a similar effect to anti-CRT antibody, which induces the upper super-shift of bands. This finding suggest that the ARE-binding protein HuR may be involved in the CRT RNA-binding complex and help CRT to interact with 3′UTR of β1-integrin mRNA. However, it needs more experiments to further confirm this observation. Other ARE-binding proteins may also contribute to the β1-integrin mRNA binding of CRT. Proteomics approaches would help to identify other binding partners of CRT in the future studies.

Although our results showed that binding of ARE at the 3′UTR is required for CRT to regulate β1-integrin mRNA stability, CRT-dependent RNA regulation might be mediated by other post-transcriptional mechanisms as well. It was reported that ARE not only locates at the 3′UTR of mRNA but also at the intron of pre-mRNA [20]. Intron plays essential roles in gene regulation, such as alternative splicing, modulation of mRNA and protein translation, and mRNA 3′-end processing [21]. Therefore, CRT may also interact with ARE at the intron of pre-mRNA, and that regulates RNA stability and protein synthesis. Likewise, GU-rich element (GRE) was identified as another highly enriched sequence in 3′UTR of mRNA, which regulates mRNA degradation [22]. Recently, CRT was reported to interact with a GRE-binding protein, CUG-binding protein 1 (CUGBP1), and suppressed the translation of C/EBPα and C/EBPβ mRNA to protein [23]. Altogether, these studies suggest that CRT may regulate mRNA stability through participating in not only ARE-binding protein complex but also GRE-binding protein complex. Therefore, whether the regulation of β1-integrin mRNA stability by CRT in PCa cells is also mediated by the GRE-dependent mechanism is worth further investigation.

In conclusion, this study demonstrated that CRT stabilizes mRNA stability through indirectly binding to ARE in the 3′UTR of mRNA. This mechanism expands our insight into the function of CRT in the regulation of mRNA stability.

## Figures and Tables

**Figure 1 biomedicines-10-00646-f001:**
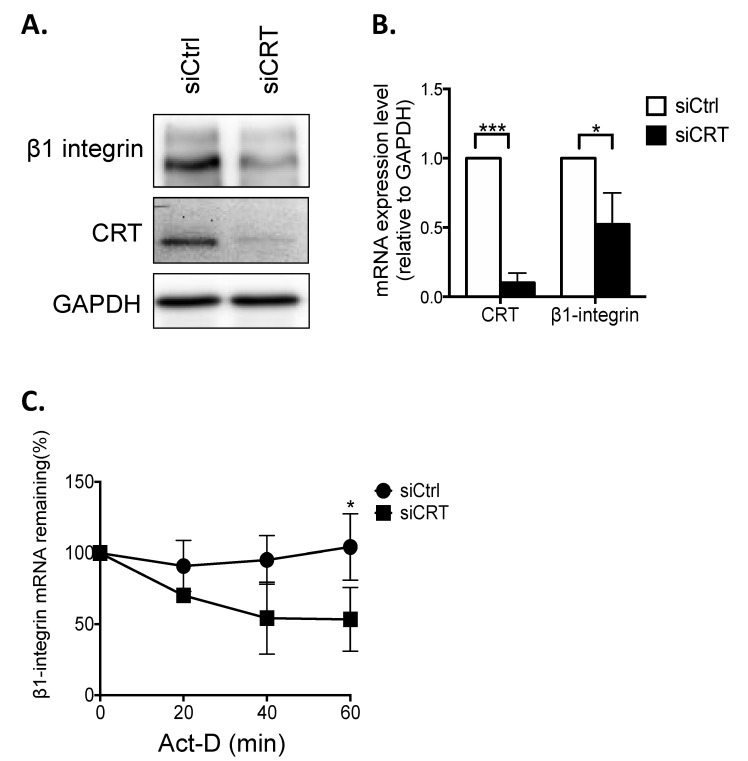
Knockdown of CRT decreases β1-integrin expression in PC-3 cells. (**A**) Western blot of CRT, β1-integrin, and FUT-1 protein expressions in PC-3 cells transfected with CRT (siCRT) and control siRNA (siCtrl) for 48 h. GAPDH was used as the internal control. (**B**) Real-time PCR analysis of mRNA expressions of CRT and β1-integrin in PC-3 cells transfected with siCRT and siCtrl for 48 h. (**C**) siCtrl and siCRT-transfected cells were treated with 2.5 µg/mL actinomycin (Act-D) for 0, 20, 40, and 60 min, and total RNA were harvested. The relative mRNA levels of CRT, β1-integrin, and VEGF-A were normalized to the internal control GAPDH. Results are shown as means ± SD. All the experiments were repeated at least three times. * *p* < 0.05; *** *p* < 0.001.

**Figure 2 biomedicines-10-00646-f002:**
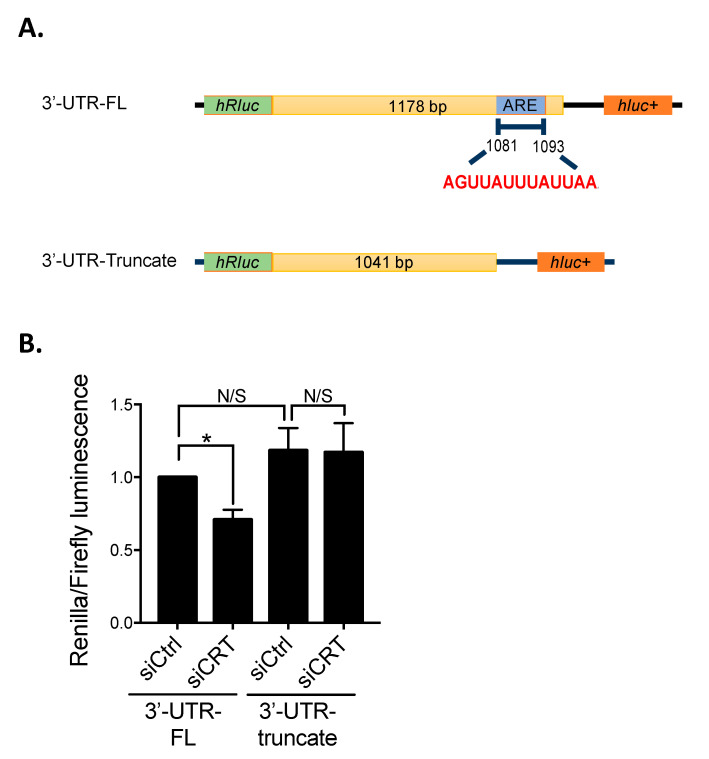
CRT stabilizes β1-integrin mRNA stability via AU-rich element (ARE) at 3′-untranslated region (3′-UTR) (**A**) Schematic diagram of psiCheck2 reporter plasmid contains firefly (hluc+) and renilla luciferase (hRluc) genes were constructed with full-length (FL) or truncated 3′UTR of β1-integrin. According to ARE online database, the sequence of ARE at 3′-UTR of β1-integrin are classified in the cluster 4. The predicted sequences are shown in red. CDS, coding sequence. (**B**) Cells were transfected with siCRT and siCtrl siRNA for 24 h, followed by transfecting with reporter plasmids for 48 h. Cotransfected cells were lysed, and cell lysates were conducted to the reporter luciferase assay. Each bar in the histogram represents the ratio of renilla luminescence to the firefly luminescence. Results are shown as mean ± SD. All the experiments were repeated at least three times. * *p* < 0.05.

**Figure 3 biomedicines-10-00646-f003:**
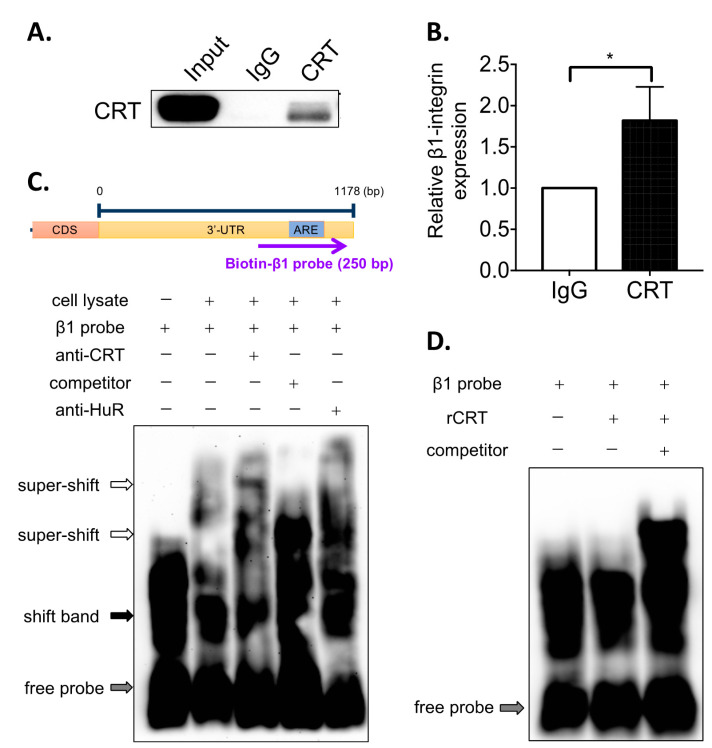
CRT interacts with β1-integrin mRNA through ARE. RNA-IP analysis of the interactions of CRT and mRNA of β1-integrin in the total cell lysate of PC-3 cells. The total cell lysates were mixed with anti-CRT antibody or anti-IgG to pull-down CRT and CRT-associated RNA. Protein-associated RNA was pulled down by magnetic protein A/G beads. (**A**) The levels of protein expressions are analyzed by Western blot. (**B**) β1 integrin RNA pulled down by anti-CRT and anti-IgG antibodies were extracted and further analyzed by real-time PCR. (**C**,**D**) RNA electrophoretic mobility shift assay (EMSA) was performed. The cell lysate was mixed with biotin-labeled ARE probes of β1-integrin and further incubated with unlabeled competitor probes or anti-CRT and anti-HuR antibodies. (**D**) Recombinant protein of CRT (rCRT, 240 ng) was mixed with biotin-labeled ARE probes of β1-integrin and further incubated with unlabeled competitor probes. Results are shown as mean ± SD. All the experiments were repeated at least three times. * *p* < 0.05.

**Figure 4 biomedicines-10-00646-f004:**
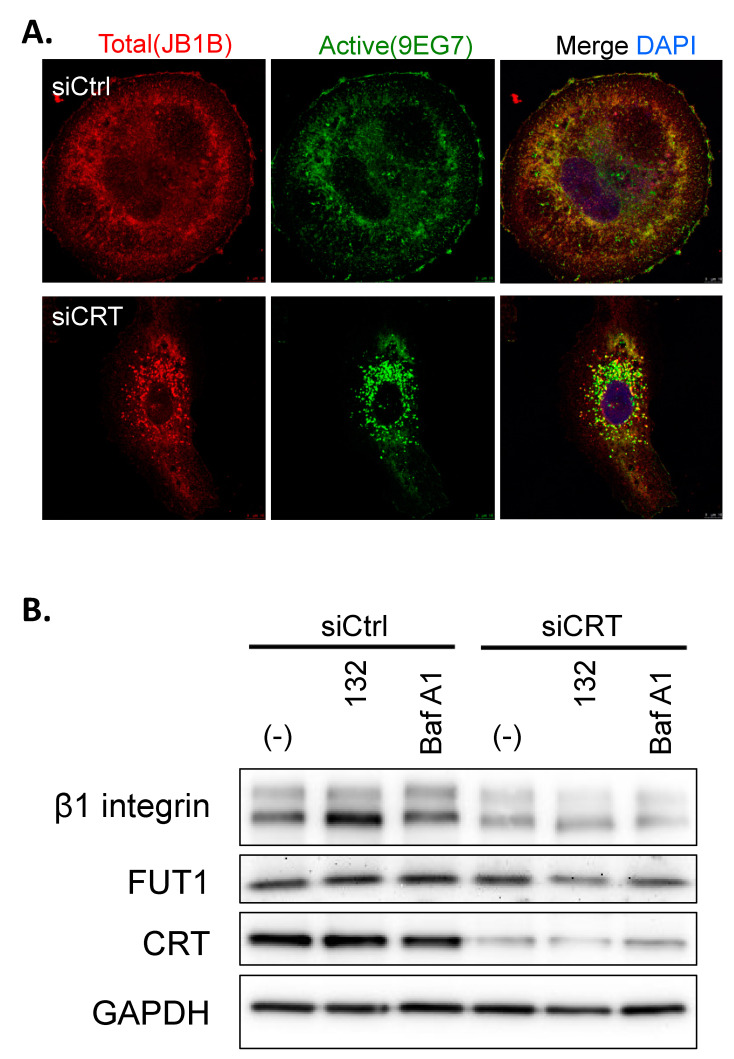
Knockdown of CRT induces β1-integrin internalization. (**A**) Confocal images of total (JB1B, red) and active (9EG7, green) β1-integrins in PC-3 transfected with siCRT or siCtrl. DAPI (blue) for nuclear staining. Scale bar = 5 μm. (**B**) PC-3 cells were transfected with siCRT- or siCtrl for 36 h and then treated with protease inhibitor MG132 (10 μM) or lysosome inhibitor Bafilomycin A1 (10 μM) for 12 h. The expression levels of β1-integrin, CRT, and GAPDH in PC-3 cell lysates were shown in Western blot images.

**Figure 5 biomedicines-10-00646-f005:**
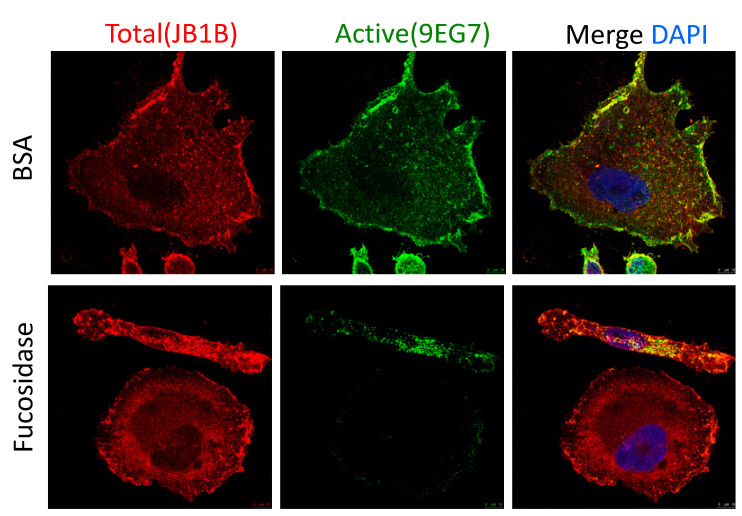
Fucosidase suppresses β1-integrin activation. Confocal images of cells treated with fucosidase or BSA for 12 h. Total (red) and activated (green) β1-integrins were shown. DAPI (blue) was used for nuclear counterstain.

**Figure 6 biomedicines-10-00646-f006:**
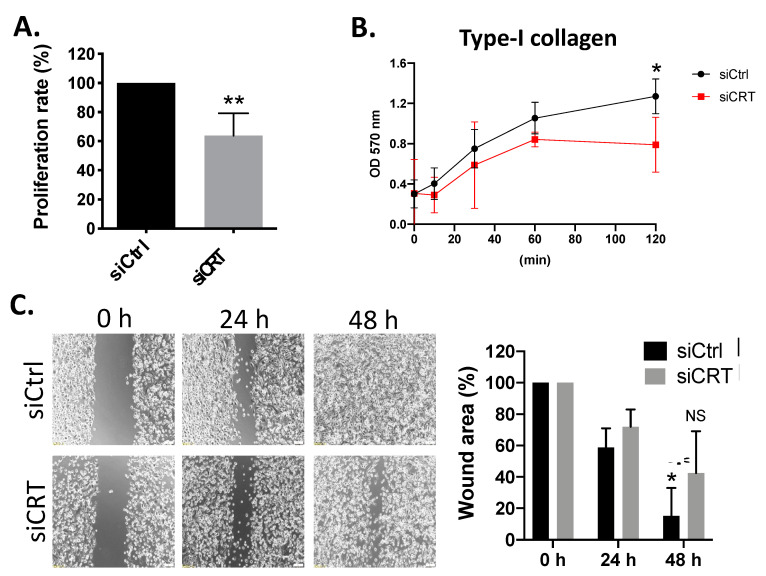
CRT regulates cell proliferation, adhesion and cell migration in PC-3 cells. (**A**) Total cell numbers of siCRT- and siCtrl-transfected PC-3 cells were counted and analyzed. (**B**) Type I collagen-mediated cell adhesions were shown. siCRT- and siCtrl-transfected PC-3 cells were plated on Type-I collagen-coated plate for 0, 10, 30, 60, and 120 min. (**C**) Representative images from wound-healing assays performed with cells transfected with siCRT or siCtrl. Confluent cell monolayers were imaged at 0, 24, and 48 h after wound scratching. Scale bar = 100 μm. Quantification of wound area was conducted by comparing the area of the wound at 24 or 48 h to that at 0 h. Each experiment was performed in triplicate, and wound healing was calculated in 4 fields/experiments. * *p* < 0.05, ** *p* < 0.01.

## Data Availability

The data that support the findings of this study are available from the corresponding author upon reasonable request.

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
