# Peer review of "Calreticulin Regulates β1-Integrin mRNA Stability in PC-3 Prostate Cancer Cells"

_biomedicines, 2022, doi:10.3390/biomedicines10030646_

Round 1

Reviewer 1 Report

Review comments are as followed:

The authors investigated the role of calreticulin (CRT), an RNA-binding protein, in PC-3 cells. The authors found the β-integrin expression level is decreased in PC-3 cells by siCRT treatment. They showed that CRT binds to 3’UTR ARE of β-integrin mRNA to enhance its stability. This stabilization of β-integrin contributes to the aggressiveness of PC-3 cells. However, I require that several points should be corrected.

  1. (Figure 2) The authors compared between siControl and siCRT. However, the effect of 3’UTR truncation on the siControl (siControl-3’-UTR-FL vs siControl-3’-UTR-truncate should be shown to determine the role of ARE sequence on this reporter system. 
  2. (Figure 3C, D) These EMSA bands are obscure and not convincing to demonstrate their claims.
  3. (Figure 4A) This result is not convincing because the immunofluorescence signals seem to be enhanced by siCRT.
  4. (Figure 6) Rescue analysis of integrin would be necessary to show that this effect of siCRT is caused by the downregulation of integrin expression level. 

Reviewer 2 Report

In this manuscript, Authors provided evidence to support the role of calreticulin in regulating expression of β1-integrin in PC-3 prostate cancer cells. The manuscript is quite relevant and provides interesting data regarding the stabilization of mRNA by indirect binding of calreticulin.

However, I have some major comments regarding the content:

  • Does calreticulin also induce stabilization of β1-integrin mRNA in other prostate cancer cells than PC-3? The use of another cell lines (LNCaP or DU145) could straighten obtained results. Authors should explain whether this mechanism is specific to the line used or not.
  • The Materials and methods section is missing some parts. How are proliferation, adhesion, and migration analysed? No information on confocal microscopy is provided.
  • Please provide full colour images in all panels of figures 4 and 5.

Round 2

Reviewer 1 Report

The authors have satisfactorily addressed my concerns.

.

Reviewer 2 Report

The manuscript is now suitable for publication.